# Autoregressive Forecasting of the Number of Forest Fires Using an Accumulated MODIS-Based Fuel Dryness Index

Daniel José Vega-Nieva [1], Jaime Briseño-Reyes [1], Pablito-Marcelo López-Serrano [2,*], José Javier Corral-Rivas [1], Marín Pompa-García [1], María Isabel Cruz-López [3], Martin Cuahutle [3], Rainer Ressl [3], Ernesto Alvarado-Celestino [4] and Robert E. Burgan [5,†]

1   Facultad de Ciencias Forestales, Universidad Juárez del Estado de Durango, Río Papaloapan y Blvd, Durango S/N Col. Valle del Sur, Durango 34120, Mexico; danieljvn@gmail.com (D.J.V.-N.); jaime.briseno@gmail.com (J.B.-R.); jcorral@ujed.mx (J.J.C.-R.); mpgarcia@ujed.mx (M.P.-G.)
2   Instituto de Silvicultura e Industria de la Madera, Universidad Juárez del Estado de Durango, Boulevard del Guadiana 501, Ciudad Universitaria, Torre de Investigación, Durango 34120, Mexico
3   Comisión Nacional para el Conocimiento y Uso de la Biodiversidad (CONABIO), Liga Periférico-Insurgentes Sur 4903, Parques del Pedregal, Del. Tlalpan, Ciudad de Mexico 14010, Mexico; icruz@conabio.gob.mx (M.I.C.-L.); mcuahutle@conabio.gob.mx (M.C.); rainer.ressl@xolo.conabio.gob.mx (R.R.)
4   School of Environmental and Forest Sciences, University of Washington, Mailbox 352100, Seattle, WA 98195, USA; alvarado@uw.edu
5   Rocky Mountain Research Station, USDA Forest Service, 1505 Khanabad Drive, Missoula, MT 59802, USA; bobinmt5@gmail.com
*   Correspondence: p_lopez@ujed.mx
†   Retired.

**Abstract:** There is a need to convert fire danger indices into operational estimates of fire activity to support strategic fire management, particularly under climate change. Few studies have evaluated multiple accumulation times for indices that combine both dead and remotely sensed estimates of live fuel moisture, and relatively few studies have aimed at predicting fire activity from both such fuel moisture estimates and autoregressive terms of previous fires. The current study aimed at developing models to forecast the 10-day number of fires by state in Mexico, from an accumulated Fuel Dryness Index (*FDI*) and an autoregressive term from the previous 10-day observed number of fires. A period of 50 days of accumulated *FDI* (*FDI*50) provided the best results to forecast the 10-day number of fires from each state. The best predictions ($R^2$ > 0.6–0.75) were obtained in the largest states, with higher fire activity, and the lower correlations were found in small or very dry states. Autoregressive models showed good skill ($R^2$ of 0.99–0.81) to forecast *FDI*50 for the next 10 days based on previous fuel dryness observations. Maps of the expected number of fires showed potential to reproduce fire activity. Fire predictions might be enhanced with gridded weather forecasts in future studies.

**Keywords:** fire danger; hazard; fuel moisture; fire occurrence; fire activity

## 1. Introduction

Temporal predictions of fire occurrence from fuel dryness (e.g., [1–4]) are fundamental to support fire management planning (e.g., [5–7]). For example, fire activity forecasts can enable effective resource pre-positioning [8]. This is even more relevant under expected higher vegetation stress from climate change, which is expected to alter the number of fires or burned area [9,10] and increase fire season length [11,12].

The majority of the literature on fire occurrence forecasting has relied on weather variables such as temperature (e.g., [13,14]) or precipitation (e.g., [1,2]), or on the use of fire danger indices calculated from observed or forecasted weather (e.g., [15–18]). The most widely used weather-based fire danger indices include those from the Canadian Forest Fire Danger Rating System, (CFFDRS) (e.g., [14–17]) or the fire danger indices of

the National Forest Fire Danger Rating System (NFFDRS) of the USA (e.g., [2,4,19]), but several challenges in fire prediction from those metrics of fuel dryness still remain [2]. In particular, several studies have concluded that the relationships between weather indices of fuel dryness and fire activity can largely vary between regions within the areas of the USA and Canada for which those fire danger indices were developed (e.g., [20,21]). These uncertainties are more evident when attempting to apply those fire weather indices in other countries that may have different biophysical and social conditions to those where they were developed (e.g., [22,23]). Thus, it is worth stressing that the majority of the analysis of fire occurrence against fire weather indices has been performed in data-abundant countries, e.g., the United States, Canada, Australia, China, and some European countries (e.g., [24]). Comparatively, fewer studies are available for global hotspots of fire activity in countries with great biodiversity, e.g., Mexico (e.g., [25,26]) or Latin American countries (e.g., [27,28]).

Furthermore, there is a need in the literature to better understand the mechanisms by which various timescales of drought are empirically related to fire occurrence [2]. While some studies have documented relationships of fire activity against weather-based indices of short-term drought (e.g., [29–31]), others have found the value of metrics of accumulated fuel dryness that account for the effect of accumulated soil and vegetation stress for weeks, or even months, to predict fire occurrence and behavior (e.g., [32–35]). In this regard, there is still a relative scarcity of studies that have compared fire danger indices at different time scales to predict fire activity (e.g., [3,36,37]). For example, Abatzoglou et al. [3] evaluated the temporal averaging of weather-based fire danger indices at periods of 1–150 days. They observed that biophysical variables tied to the depletion of fuel and soil moisture and prolonged periods of elevated fire danger had stronger correlations to areas burned in forested systems. Riley et al. [2] found that metrics based on the previous 1–3 months of weather data had strong correlations with both the total burned area and the number of large fires. Gudmundsson et al. [36] and Turco et al. [36] evaluated different periods of the Standardized Precipitation Index (SPI) index to predict burned area. They found the highest correlations for 2- and 3-month SPI, respectively. An improved understanding of the time scales through which weather influences fire occurrence could be beneficial to support operational fire management, particularly under climate change [2].

Weather-based fire indices do not explicitly include vegetation vulnerability to fires due to drought. Nevertheless, the flammability of live fuels depends not only on weather variability, but also on plant and soil response to it, which is species- and landscape-specific (e.g., [38–40]). Although live fuel moisture has been shown to be crucial in predicting fire behavior and fire severity (e.g., [39–42]), its role is not explicitly accounted for by current fire danger models [43]. In this sense, in spite of an emerging body of literature that has demonstrated a correlation of remotely sensed estimates of live moisture (e.g., [44,45]) with fire occurrence and behavior (e.g., [46–48]), there are still uncertainties for incorporating such remotely sensed fuel moisture estimates into operational fire prediction modeling (e.g., [38,39,42]).

Further, some studies have proposed an integration of MODIS relative greenness and dead fuel moisture on a Fire Potential Index [49] at 1 km resolution, with promising results in the USA (e.g., [50,51]), Europe ([52–54]), and, more recently, Mexico [26]. Even though FPI has shown a good potential to predict fire occurrence (e.g., [51]), further research evaluating this index at different time lags is lacking. In particular, we are not aware of studies that have compared different accumulation times (e.g., within the range of 10 to 90 days) for the FPI index.

Moreover, except for the studies of Huesca et al. [53,54], the use of autoregressive models to forecast the FPI index has received little attention in the literature, in spite of its potential to estimate future fuel greenness conditions based on previously observed remotely sensed fuel moisture.

Finally, beyond considering the effects of fuel moisture in allowing fire initiation, the occurrence of forest fires is largely conditioned by the occurrence of an ignition (e.g., [8]). In the case of natural-caused fires, ignitions can be caused by lightning (e.g., [14,55,56]).

Human-caused fires, on the other hand, can be related to spatial and temporal patterns of human activities such as escaped agricultural burns or urban or agricultural expansion (e.g., [57–60]). Regarding the latter, several studies have documented that anthropogenic factors can largely influence fire occurrence (e.g., [61,62]), particularly in regions with frequent conversion to agricultural lands (e.g., [63]). In the case of Mexico, the statistics by the National Forest Commission (CONAFOR) [64] show that more than 90% of suppressed fires are related to human activities.

In this regard, in order to account for the important role of human-based temporal patterns on fire activity, some studies have suggested a good potential for autoregressive techniques (e.g., [65,66]). Compared to a vast majority of studies predicting fire activity from fuel dryness only, this promising autoregressive approach for temporal fire forecasting has nevertheless received relatively less attention in the literature (e.g., [67,68]) and demands further research. In particular, there is a relative knowledge gap in studies aiming at predicting fire activity from both autoregressive fire activity and fuel moisture (e.g., [69–71]).

Consequently, the aim of this study is to develop models to forecast the number of fires by state in Mexico from both autoregressive fire activity and accumulated fuel moisture. In particular, the specific objectives of the study were as follows:

(1) To develop models to predict the number of fires by state in Mexico for the next 10 days from an accumulated Fuel Dryness Index (*AcFDI*) and autoregressive terms from the number of fires observed in the previous 10 days;

(2) To develop autoregressive models to forecast the *AcFDI* for the next 10 days by state in Mexico.

## 2. Materials and Methods

### 2.1. Study Area and Fire Suppression Records

The study area was the entire country of Mexico (Figure 1). Main vegetation types range from desert shrublands and temperate forests in the states of the north to tropical forests in the south of the country [72]. Precipitation ranges from <500 mm in the more arid states of the north to >1000 mm in the tropical southern region [73].

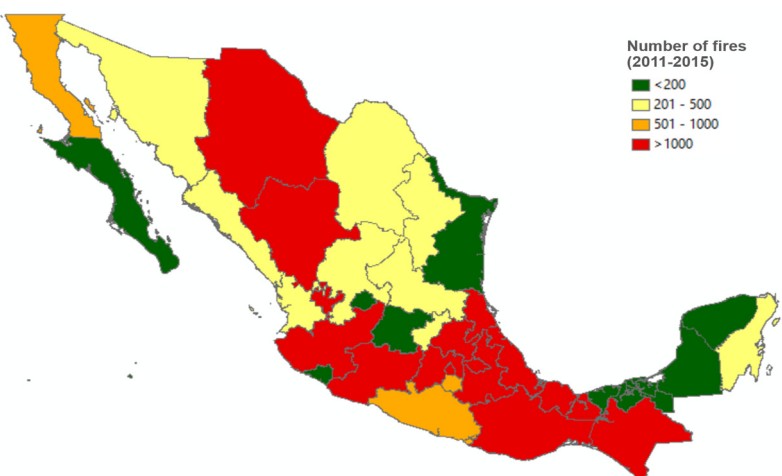

**Figure 1.** Number of suppressed fires by state in the study period (2011–2015).

We analyzed fire suppression records from CONAFOR [64] from 2011 to 2015 for every state of Mexico. The database contains the start and end date of each fire, based on fire suppression records, their coordinates, and the corresponding state of Mexico. The total number of fire suppression records in the study period was 38,715. The states with a higher number of fire records included State of Mexico, Chihuahua, Durango, Michoacan, Ciudad de Mexico, Jalisco, Puebla, Chiapas, and Oaxaca, with more than 1000 fire suppression records for the study period (Figure 1). Conversely, the states with the lower number of

fire registers were Aguascalientes, Baja California Sur, Colima, Campeche, Guanajuato, Tamaulipas, Tabasco, and Yucatan (Figure 1).

### 2.2. Fuel Dryness Index

#### 2.2.1. Inputs for the Fuel Dryness Index (*FDI*) Calculation

The two inputs for *FDI* calculation, moisture content of dead fuels of 100 h (*H*100) and 10-day *NDVI* composites, were supplied by the National Commission for the Knowledge and Use of Biodiversity (CONABIO), as detailed by Cruz-Lopez et al. [74]. H100 composites at 1 km pixel were calculated by CONABIO using the methodology of Cervera-Tavoada [75] to implement the NFDRS of dead fuel moisture content calculation [76] from MODIS temperature and relative humidity and TRMM precipitation [74]. The period of study was 2011–2015, defined by the availability of *H*100 data from CONABIO at the time of analysis.

The 10-day cloud-free *NDVI* composite images, with a pixel of 1 km, were calculated from MODIS by CONABIO [74]. The process of gap filling to reconstruct cloud-free *NDVI* composites was performed by CONABIO using the HANTS (Harmonic Analysis of Time Series) algorithm [77]. HANTS is a widely used algorithm to reconstruct time series for seasonal stationary variables such as *NDVI* [78]. The algorithm was applied to a time series of 9 years of MODIS *NDVI* to obtain a fitted time series (using a superposition of periodic functions [77]) to reconstruct the temporal gaps for each pixel.

#### 2.2.2. Fuel Dryness Index (*FDI*) Calculation

The Fuel Dryness Index (*FDI*) was calculated using the procedure described by [26] to calibrate the Fire Potential Index (*FPI*) from Burgan et al. [49] for Mexican ecosystems:

$$FDI = (1 - LR) \cdot (1 - MR) \cdot 100 \tag{1}$$

where:

*LR:* Live ratio calculated using Equations (2)–(4).

*MR*: Dead fuels moisture ratio calculated using Equation (5).

*FDI* is an integrated index that combines both estimates of live fuel moisture (LR) (Equation (1) through (4)) and dead fuel Moisture Ratio (*MR*) (Equations (1) and (5)), with a spatial resolution of 1 km. It reaches values close to 100 when the pixel reaches its maximum fuel dryness (minimum dead and fuel moisture). Conversely, its lowest values are reached when both live and dead fuels have high moisture [49].

The first component, Live Ratio (*LR*), was calculated using Equations (2) and (3):

$$LR = RG \cdot LRmax / 100 \tag{2}$$

where: *RG* is Relative Greenness, estimated as:

$$RG = (NDVI - NDVImin) / (NDVImax - NDVImin) \cdot 100 \tag{3}$$

where: *NDVI* is the observed 10-day *NDVI* for each pixel, *NDVImin* and *NDVImax* are the minimum and maximum *NDVI* values for each pixel from the period of study.

*LRmax* is the maximum Live Ratio value for each pixel, calculated using Equation (4):

$$LRmax = 30 + 30 \cdot (NDVImax - 125) / (255 - 125) \tag{4}$$

where: *NDVImax* = maximum *NDVI* for every pixel.

The values of *NDVI* were scaled from 0 to 255 to permit data compression, as detailed in [79]. The map of the maximum *NDVI* observed for each pixel in the study period ranged from 125 to 255 [26]. Following [79], the absolute minimum (125) and maximum (255) values of the maximum *NDVI* were included in Equation (4).

The value of 30, at the intercept of Equation (4), represents the minimum *LRmax* value, as established by [79]. Following Equation (4), a maximum *LRmax* value of 60 is reached for the pixels where the maximum *NDVI* value reaches its absolute observed value for

the study area. Consequently, as proposed for *FPI* [61,79], areas with a lower maximum *NDVI* (e.g., desert shrublands) have a lower maximum live ratio (fraction of fuels that is estimated to be alive) and the contrary occurs in areas with higher *NDVImax*.

Finally, the dead fuel Moisture Ratio (*MR*) was calculated following Equation (5):

$$MR = (H100 - Hmin)/(Hmax - Hmin) \tag{5}$$

where: *H*100: observed 100 h dead fuel moisture; *Hmax*, *Hmin*: maximum and minimum historical *H*100 values for each pixel.

### 2.2.3. Accumulated *FDI*

We tested accumulated *FDI* (*AcFDIi*), calculated as the average *FDI* value for the evaluated *i* periods of 10, 20, 30, 40, 50, 60, 70, 80, and 90 days, for every Mexico state. Evaluated periods of 10–90 days for the *AcFDIi* were selected based on the more common range of accumulated periods for fire danger indices considered in the literature (e.g., [17,18]). The index *AcFDI* was assumed to be zero when the FDI value at the corresponding 10-day period was below a threshold $FDI_{99}$. For each state, $FDI_{99}$ thresholds were calculated as the *FDI* value above which 99% of the fire suppression registers were registered. The average *AcFDI* was calculated for each state for every period of 10 days in the study period.

### 2.3. Models for Prediction of Number of Fires for Each State

We evaluated linear and non-linear models to predict the observed number of fires for 10 days for each state from *AcFDI* and the observed number of fires in the last 10 days. For the non-linear models, we fitted the following expression [26]:

$$NFt = a \cdot AcFDIit^{b} + c \cdot NFt - 1 \tag{6}$$

where: *NFt*: Observed number of fires for each state for each 10-day period *t* of the study period; *AcFDIit*: Accumulated Fuel Dryness Index at each 10-day period *t* of the study period for the evaluated accumulated period *i* (Section 2.2.3); *NFt* − 1: Observed number of fires for each state for the previous 10-day period *t* − 1; *a*, *b*, are model coefficients for the role of *AcFDI*, fitted using non-linear quantile regression at a 95% percentile, and *c* is a model coefficient to account for the temporal autocorrelation of *NF*, which was estimated as the calculated correlation coefficient between observed values of *NFt* and *NFt* − 1 for each state following [26].

For obtaining the *a* and *b* model coefficients of *AcFDI* for each state, we fitted the models from Equation (6) using non-linear quantile regression at a 95% percentile using the R package *nlrq* [80]. Candidate models were evaluated by means of the coefficient of determination for non-linear regression ($R^2$) (e.g., [81]), defined as the squared correlation coefficient between the measured and estimated values, together with the Root Mean Square Error (RMSE) and model bias.

### 2.4. Autoregressive Integrated Moving Average (ARIMA) Models to Forecast AcFDI

For the selected *AcFDIi* index, we evaluated seasonal AutoRegressive (AR) Integrated (I) Moving Average (MA) models (ARIMA) to forecast the *AcFDIi* for the next 10 days for each state, based on previously observed lags of the same index. Seasonal ARIMAs are frequently used to forecast time series of remotely sensed estimates of fuel moisture such as NDVI (e.g., [82]). They have been previously used to model FPI temporal dynamics by Huesca et al. [53,54].

A generic notation of ARIMA models can be written as:

$$ARIMA\ (ar, dif, ma) \times (sar, sdif, sma)S, \tag{7}$$

where: *ar* = non-seasonal AR lag order, *dif* = non-seasonal differencing, *ma* = non-seasonal MA lag order, *sar* = seasonal AR lag order, *sdif* = seasonal differencing, sma = seasonal MA lag order, and S = time span of repeating seasonal pattern.

Lag order and components of ARIMA models were identified by an exploratory analysis using auto.arima [53] in R ([83,84]). The most suitable model was selected based on Standard AIC [85], together with the evaluation of model $R^2$, RMSE, and bias. Selected models were fitted using auto.arima in the library *forecasting* in R ([83,84]). The adequacies of selected models were further evaluated by means of the Ljung–Box Q-statistic [53,86].

For example, if auto.arima selected a (2, 0, 0) × (0, 0, 0)S as the best model, which is an autoregressive model to predict *AcFDI* from the two previous lags, the corresponding AR model can be written as:

$$AcFDI = a0 + a1 \cdot AR_1 + a2 \cdot AR_2 \tag{8}$$

where $AR_1$ and $AR_2$ are the autoregressive *AcFDI* observed values for lag 1 (i.e., previous 10 days) and lag 2 (previous 11–20 days), respectively, and *a*0, *a*1, and *a*2 are fitted model coefficients using auto.arima ([83,84]).

## 3. Results

### 3.1. FDI<sub>99</sub> Thresholds by State

An example of the process of calculation of $FDI_{99}$ is illustrated in Figure 2. The upper Figure 2a shows the curves of the accumulated percentage of the number of suppressed fires against *FDI* values. For illustration purposes, we show a selected example for the states of Oaxaca, Durango, and state of Mexico. A detail of the same curves, for the accumulated percentages of fires below 10%, is shown in Figure 2b. $FDI_{99}$ values, obtained as the nearest *FDI* integer for the accumulated % fires for the first 1% (i.e., 99% of fires occur above this *FDI*), are marked in circles for each curve in Figure 2b. $FDI_{99}$ values of 40, 48, and 55 were obtained for the states of Oaxaca, Mexico, and Durango, respectively (Figure 2b).

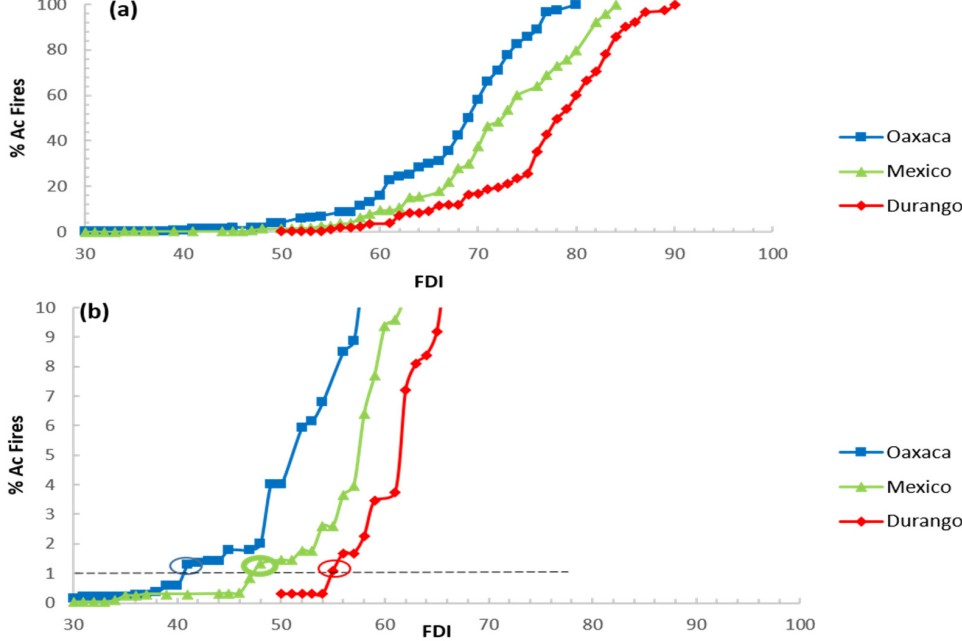

**Figure 2.** Accumulated percentage of fires (% Ac Fires) against Fuel Dryness Index (FDI) values for the states of Oaxaca (blue), Durango (red), and State of Mexico (green) (**a**) and detail of the same curves for the % Ac Fires below 10% (**b**). $FDI_{99}$ values (*FDI* values corresponding to a % Ac Fires of 1) are marked as circles for each curve.

The observed $FDI_{99}$ values for all the states ranged from 33 to 60 (Figure 3, Table 1). The lowest $FDI_{99}$ values (<40) were observed in some of the wetter states of the southeast, such as Yucatan, Chiapas, or Oaxaca. On the contrary, higher $FDI_{99}$ values (>50) were

observed for the driest states in the Northern region (e.g., Durango, Chihuahua), or Center (e.g., Hidalgo, Guanajuato, San Luis Potosi).

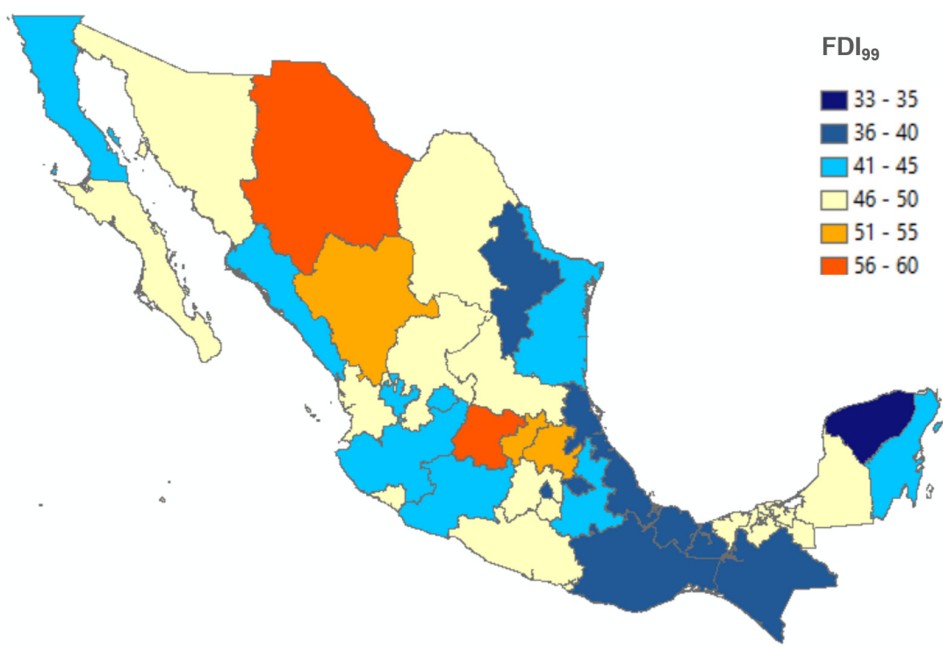

**Figure 3.** Map of $FDI_{99}$ values by state of Mexico.

**Table 1.** Coefficient and goodness of fit of models to predict number of fires in 10 days by state.

| State | $FDI_{99}$ | a | b | c | $R^2$ | RMSE | Bias |
|---|---|---|---|---|---|---|---|
| Baja California | 45 | 34.701 | 3.227 | 0.665 | 0.501 | 6.840 | −4.160 |
| Campeche | 47 | 27.366 | 2.409 | 0.400 | 0.450 | 1.689 | −0.379 |
| Chiapas | 38 | 103.179 | 3.230 | 0.733 | 0.631 | 11.889 | −6.297 |
| Chihuahua | 58 | 280.047 | 6.273 | 0.819 | 0.743 | 40.310 | −20.510 |
| Coahuila | 48 | 36.343 | 3.696 | 0.726 | 0.492 | 5.228 | −2.360 |
| Colima | 48 | 56.733 | 3.634 | 0.332 | 0.160 | 9.916 | −7.684 |
| Ciudad de Mexico | 39 | 645.171 | 7.069 | 0.732 | 0.625 | 30.580 | −12.710 |
| Durango | 54 | 99.630 | 5.130 | 0.686 | 0.559 | 13.700 | −5.647 |
| Guanajuato | 60 | 24.410 | 4.219 | 0.168 | 0.178 | 6.308 | −5.020 |
| Guerrero | 47 | 25.485 | 0.987 | 0.587 | 0.377 | 8.945 | −4.716 |
| Hidalgo | 53 | 78.719 | 2.699 | 0.642 | 0.466 | 15.061 | −7.640 |
| Jalisco | 41 | 77.550 | 2.736 | 0.847 | 0.673 | 21.012 | −12.265 |
| Mexico | 48 | 654.915 | 6.053 | 0.730 | 0.628 | 48.452 | −19.110 |
| Michoacan | 41 | 276.791 | 5.361 | 0.767 | 0.646 | 34.615 | −15.674 |
| Morelos | 50 | 37.675 | 2.794 | 0.514 | 0.379 | 8.480 | −4.510 |
| Nayarit | 46 | 12.851 | 0.970 | 0.730 | 0.554 | 4.562 | −2.198 |
| Nuevo Leon | 39 | 17.617 | 1.960 | 0.726 | 0.531 | 4.297 | −1.980 |
| Oaxaca | 40 | 99.594 | 4.216 | 0.702 | 0.574 | 10.100 | −5.110 |
| Puebla | 43 | 161.332 | 4.319 | 0.566 | 0.479 | 20.270 | −10.778 |
| Queretaro | 55 | 102.881 | 7.392 | 0.321 | 0.257 | 5.379 | −2.137 |
| Quintana Roo | 43 | 25.036 | 1.517 | 0.646 | 0.471 | 4.210 | −1.711 |
| San Luis Potosi | 50 | 107.705 | 8.867 | 0.458 | 0.279 | 4.221 | −1.367 |
| Sinaloa | 45 | 33.935 | 4.439 | 0.213 | 0.229 | 3.567 | −1.484 |
| Sonora | 50 | 24.221 | 3.069 | 0.350 | 0.253 | 4.384 | −2.047 |
| Tabasco | 47 | 25.034 | 1.517 | 0.646 | 0.479 | 4.291 | −1.770 |
| Tamaulipas | 45 | 25.469 | 3.402 | 0.247 | 0.153 | 1.310 | −0.412 |
| Tlaxcala | 40 | 90.834 | 3.587 | 0.652 | 0.537 | 14.450 | −7.920 |
| Veracruz | 39 | 150.319 | 3.854 | 0.612 | 0.411 | 13.598 | −7.550 |
| Yucatan | 35 | 109.077 | 5.544 | 0.294 | 0.147 | 4.115 | −1.468 |

Where: $FDI_{99}$: FDI values corresponding to a % Ac Fires of 1%; *a*, *b*, and *c* are model coefficients (Equation (6)); $R^2$: coefficient of determination; RMSE: Root Mean Squared Error.

### 3.2. Models to Predict Number of Fires by State

Based on a preliminary correlation analysis between the candidate accumulated *FDI* for the evaluated periods of 10–90 days and the observed number of fires, a period of 50 days was selected for the accumulated Fuel Dryness Index (*FDI*50) based on an observed higher correlation to predict fire activity for the majority of the analyzed states. The best fits were obtained using non-linear models (Equation (6)). The coefficients and goodness of fit for the models to predict the number of fires from *FDI*50 using Equation (6) are shown in Table 1. For 19 of the 32 evaluated states, values of $R^2$ higher than 0.4 were observed (Table 1).

In general, the best performance ($R^2$ of up 0.6–0.75) was observed in states with higher fire activity (states with >1000 fires in the study period, Figure 1), such as Chihuahua, Ciudad de Mexico, Jalisco, Michoacan, or the state of Mexico. Conversely, the lowest goodness of fit was observed for states with a low number of fire suppression records (<200 fires in Figure 1), such as Colima, Guanajuato, Tamaulipas, or Yucatan (Table 1). For the two states with very low fire activity, Southern Baja California and Aguascalientes, $R^2$ values lower than 0.1 (models not shown) were obtained. The fitted models from the nearest states (Baja California and Zacatecas, respectively), scaled by the corresponding forest land surface, were applied for those two states. Because of the use of percentile regression, all models showed negative bias values (i.e., the models rarely provide underestimations, as required for risk assessment).

Selected examples of predicted against the observed number of fires in 10 days are shown in Figure 4 for the states of Chihuahua (North region), Mexico (Centre), and Oaxaca (South region). In general, the models provide conservative predictions, with relatively few underestimates, because of the use of percentile regression (to represent "worst case scenarios"), combined with the consideration of autocorrelation, that corrects for unexpected punctual events (i.e., high observed fire activity under relatively wet conditions), that are considered in the prediction for the next 10 days. Additional examples of predicted against observed number of fires are shown in Supplementary Figures S1–S5.

### 3.3. Autoregressive Models to Forecast Accumulated Fuel Dryness

Based on the exploratory analysis using *autoarima*, autoregressive models of order 2, i.e., (2, 0, 0) × (0, 0, 0)S (Equation (8)), were selected. The coefficients for the selected models using Equation (8) to forecast *FDI*50 are shown in Table 2. $R^2$ ranged from 0.989 to 0.813 and RMSE from 1.70 to 7.54. For 29 out of the 32 evaluated states, $R^2$ values were higher than 0.95 and RMSE values were lower than 2.5. In addition, all the fitted models demonstrated a lack of autocorrelation in the residues based on the Box–Ljung test.

**Table 2.** Coefficient and goodness of fit of autoregressive models to forecast accumulated fuel dryness *FDI*50.

| State | $a0$ | $a1$ | $a2$ | $R^2$ | RMSE | Bias | AIC |
|---|---|---|---|---|---|---|---|
| Aguascalientes | 2.142 | 1.720 | −0.756 | 0.982 | 2.440 | 0.021 | 803.76 |
| Baja California | 7.582 | 1.386 | −0.503 | 0.888 | 2.389 | −0.119 | 812.47 |
| Baja California Sur | 2.620 | 1.498 | −0.538 | 0.961 | 2.219 | 0.024 | 769.92 |
| Campeche | 2.989 | 1.676 | −0.739 | 0.971 | 1.703 | 0.034 | 701.53 |
| Coahuila | 2.007 | 1.430 | −0.462 | 0.964 | 2.061 | 0.037 | 761.53 |
| Colima | 2.093 | 1.781 | −0.822 | 0.813 | 7.542 | 0.239 | 1193.45 |
| Chiapas | 2.364 | 1.742 | −0.793 | 0.981 | 1.807 | 0.028 | 720.09 |
| Chihuahua | 2.728 | 1.751 | −0.794 | 0.982 | 2.035 | 0.011 | 758.62 |
| Ciudad de Mexico | 2.196 | 1.632 | −0.676 | 0.973 | 2.608 | 0.037 | 845.13 |
| Durango | 2.266 | 1.759 | −0.798 | 0.985 | 2.086 | 0.012 | 767.30 |
| Guanajuato | 2.075 | 1.689 | −0.727 | 0.985 | 2.150 | 0.052 | 781.91 |
| Guerrero | 2.132 | 1.828 | −0.869 | 0.989 | 1.855 | 0.004 | 726.84 |
| Hidalgo | 2.070 | 1.635 | −0.675 | 0.975 | 2.340 | 0.035 | 807.23 |
| Jalisco | 2.061 | 1.813 | −0.853 | 0.989 | 2.025 | 0.005 | 757.53 |

**Table 2.** *Cont.*

| State | a0 | a1 | a2 | R² | RMSE | Bias | AIC |
|---|---|---|---|---|---|---|---|
| Mexico | 1.794 | 1.768 | −0.807 | 0.985 | 2.331 | 0.047 | 811.73 |
| Michoacan | 1.847 | 1.816 | −0.853 | 0.989 | 2.006 | 0.036 | 759.73 |
| Morelos | 2.058 | 1.778 | −0.818 | 0.986 | 2.428 | 0.013 | 821.03 |
| Nayarit | 2.132 | 1.834 | −0.876 | 0.988 | 1.983 | 0.004 | 750.37 |
| Nuevo Leon | 1.682 | 1.405 | −0.434 | 0.975 | 1.840 | 0.040 | 737.65 |
| Oaxaca | 2.239 | 1.742 | −0.787 | 0.984 | 1.870 | 0.069 | 741.15 |
| Puebla | 2.151 | 1.706 | −0.748 | 0.982 | 2.259 | 0.070 | 802.59 |
| Queretaro | 1.991 | 1.641 | −0.679 | 0.979 | 2.290 | 0.063 | 808.12 |
| Quintana Roo | 3.630 | 1.591 | −0.669 | 0.952 | 1.799 | 0.033 | 719.08 |
| San Luis Potosi | 1.487 | 1.630 | −0.657 | 0.976 | 2.181 | 0.007 | 792.53 |
| Sinaloa | 3.092 | 1.822 | −0.875 | 0.964 | 1.820 | 0.002 | 718.40 |
| Sonora | 4.340 | 1.744 | −0.811 | 0.947 | 2.167 | 0.002 | 780.40 |
| Tabasco | 3.425 | 1.608 | −0.678 | 0.955 | 2.117 | 0.024 | 771.23 |
| Tamaulipas | 2.097 | 1.512 | −0.549 | 0.965 | 1.926 | 0.043 | 737.93 |
| Tlaxcala | 2.044 | 1.686 | −0.728 | 0.979 | 2.903 | 0.042 | 883.19 |
| Veracruz | 2.295 | 1.675 | −0.722 | 0.974 | 1.774 | 0.016 | 709.54 |
| Yucatan | 2.386 | 1.664 | −0.714 | 0.971 | 1.981 | 0.000 | 748.31 |
| Zacatecas | 1.668 | 1.707 | −0.736 | 0.982 | 2.339 | 0.070 | 818.07 |

Where: $a0$, $a1$, and $a2$ are model coefficients for Equation (7). $R^2$: coefficient of determination; RMSE: Root Mean Squared Error.

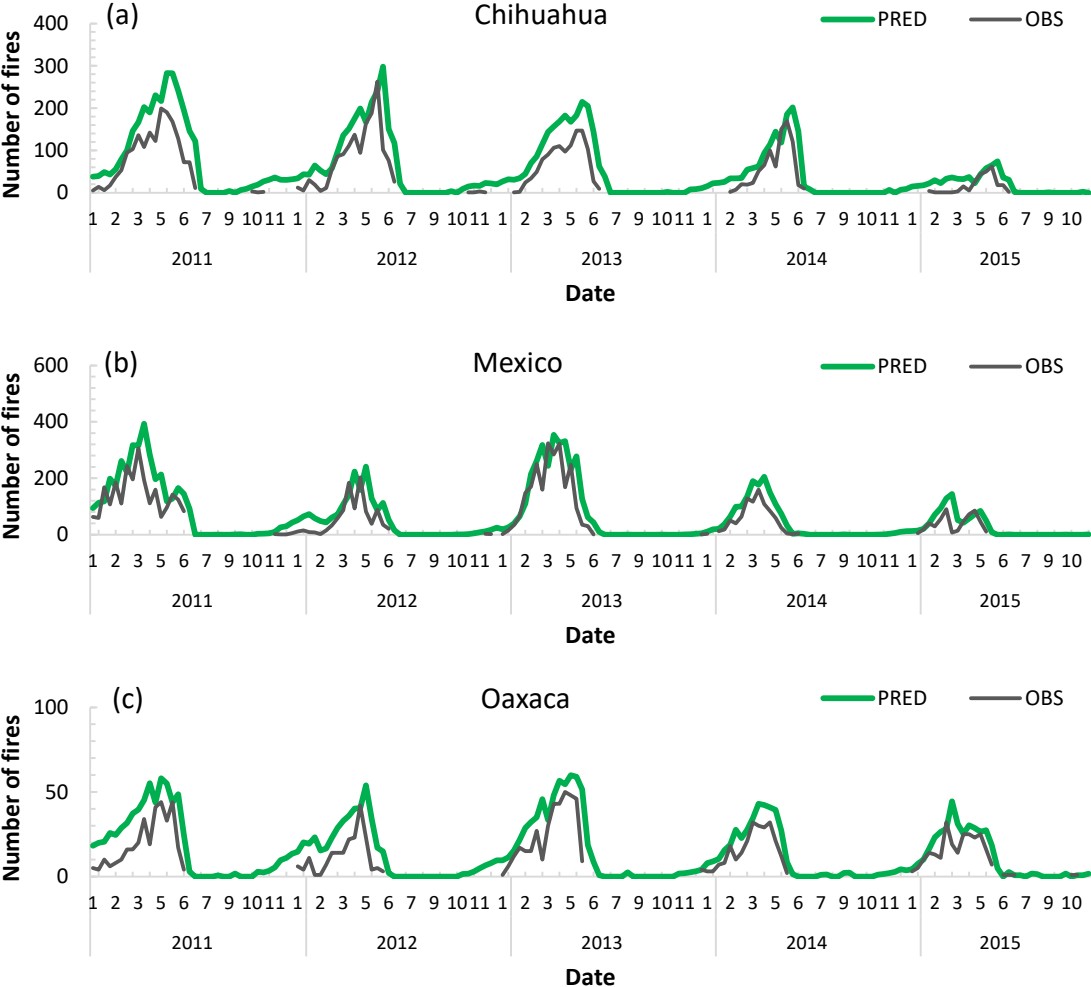

**Figure 4.** Selected examples of observed and predicted number of fires in 10 days for the states of Chihuahua (**a**), State of Mexico (**b**), and Oaxaca (**c**). Where PRED: predicted with models from Table 1; OBS: observed.

Selected examples of observed against predicted *FDI*50 values using the fitted autoregressive models are shown in Figure 5, where a close agreement between forecasted and observed fuel dryness can be observed. Plots of observed against predicted *FDI*50 values for all the evaluated states are shown in Supplementary Figures S6–S10.

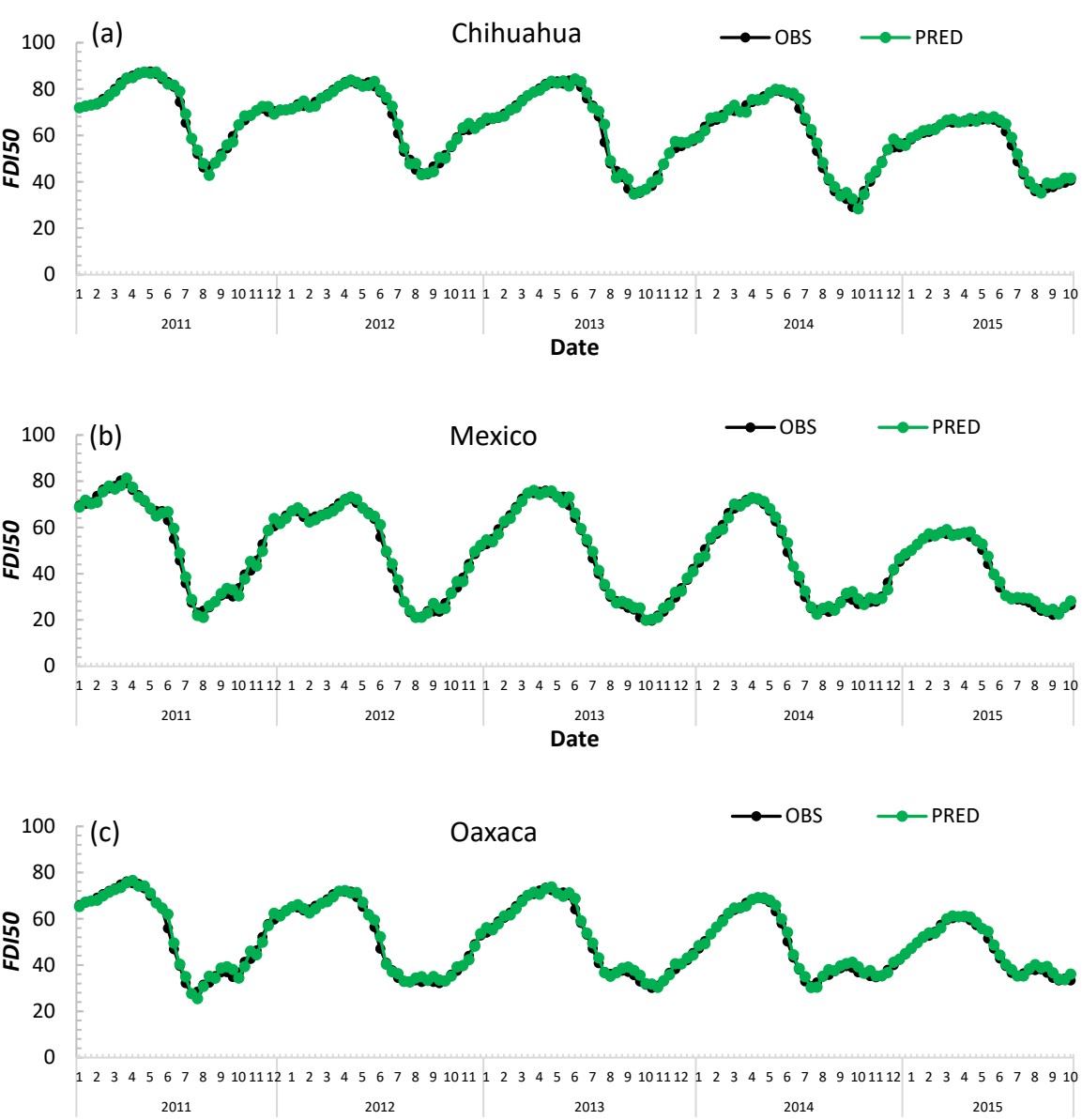

**Figure 5.** Selected examples of observed and predicted accumulated fuel dryness *FDI*50 for the states of Chihuahua (**a**), State of Mexico (**b**), and Oaxaca (**c**).

The developed models allow us to map the estimated number of fires by state for every 10-day period, based on the forecasted accumulated fuel dryness *FDI*50 and on the observed number of fires from the previous 10 days. An example of the forecasted number of fires for two contrasting years (dry year 2011 and wet year 2015) is shown in Figure 6. The forecasted number of fires shows sensitivity to both years and time periods within years of contrasting fuel dryness and previous fire activity, generally matching with observed fire suppression registers (shown as blue dots in Figure 6).

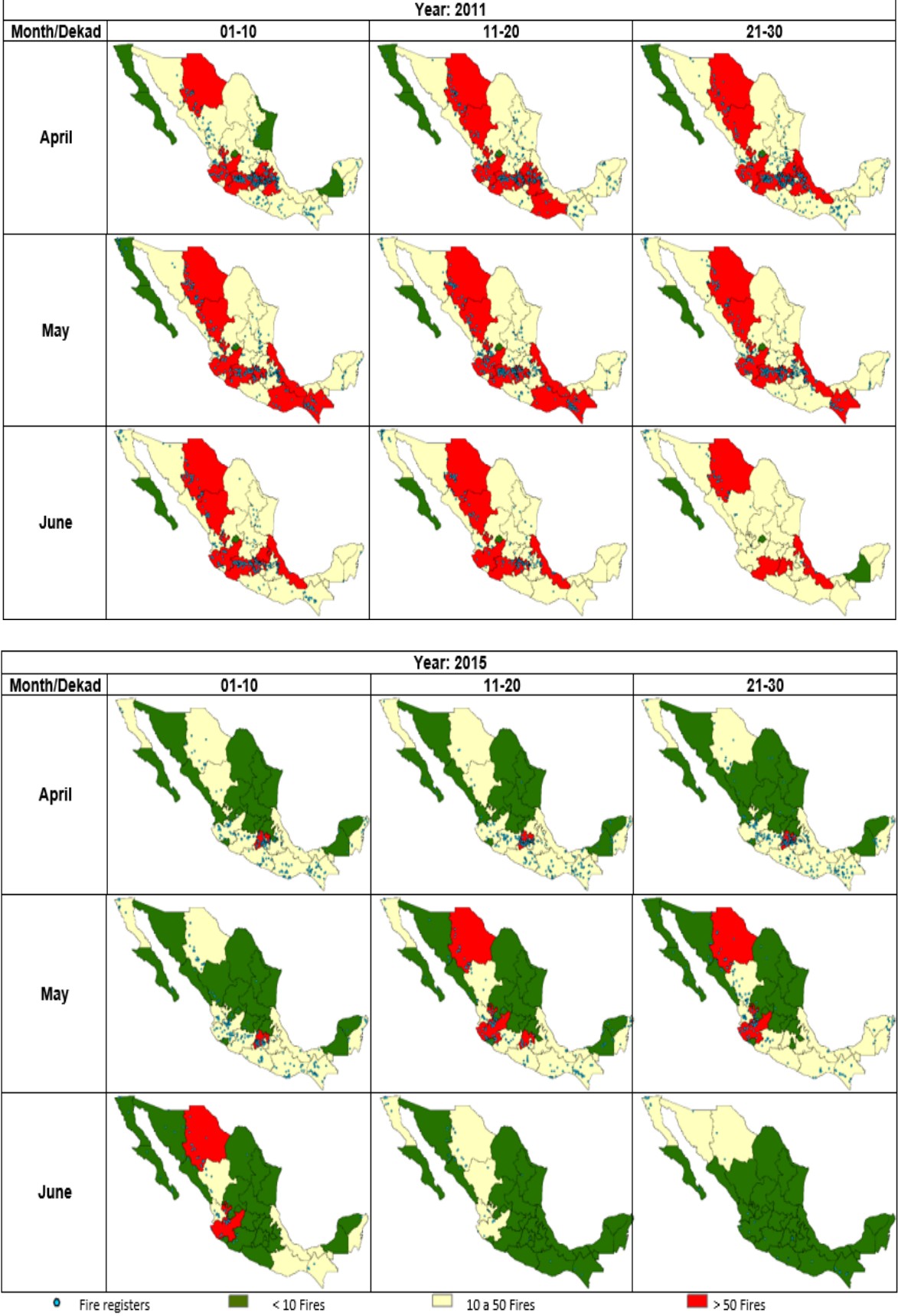

**Figure 6.** Predicted number of fires by state for the period April–June of 2011 (three upper rows) and 2015 (three lower rows). Observed fire suppression registers are shown as blue dots in each figure.

## 4. Discussion

This study demonstrated the potential of the combination of forecasts of remotely sensed fuel moisture, together with autoregressive models that account for human-caused temporal variations in fire activity, to predict fire occurrence at temporal (10 days) and spatial units relevant for fire management planning.

Regarding the analyzed effect of fuel dryness, the observed variations between states in the $FDI_{99}$ thresholds (Figure 2) in the current study support previous observations that fuel dryness thresholds for fire occurrence can vary between geographical regions (e.g., [87]). Furthermore, the generally lower thresholds for fire occurrence observed in wetter states and, conversely, higher $FDI_{99}$ found for drier states, support similar previous observations for *FPI* by [60], who documented the highest *FPI* thresholds for the drier bioclimatic regions in Southern Europe. This also agrees with previous observations of higher thresholds of live fuel moisture for fire occurrence in drier ecosystems than in wetter, more productive forests (e.g., [40,43,88]).

For several states, the developed models showed some skill in relating the higher observed fire activity (e.g., years 2011–2013 for Chihuahua, Figure 4a) to corresponding higher accumulated fuel dryness in those periods (corresponding years and state in Figure 4a) and, conversely, in explaining the lower fire activity observed in wetter years (e.g., year 2015 for Chihuahua or Mexico, Figures 4a and 5a). This interannual variability in fire activity has been previously documented to be related to annual ENSO indices for Mexico (e.g., [89–91]) and other countries (e.g., [92–94]), although the country- and region-specific effects of Niño/Niña are still not fully understood for many areas of Mexico (e.g., [95–97]). The temporal patterns of observed fire activity (Figure 4 and Supplementary Figures S1–S5) and fuel dryness (Figure 5 and Supplementary Figures S6–S10) point that, for several states in the analysis, particularly in the North of Mexico, higher fire activity is observed for La Niña years, such as 2011, which had been documented to be a record for both number of fires and burned area in Mexico [64].

Our selection of an accumulated fuel dryness index for prediction of fire occurrence seems to agree with studies that have observed higher correlations of fire activity with longer time lag fire weather indices (e.g., [35,98,99]), reinforcing the hypothesis that antecedent water balance and accumulated drought can influence fire activity (e.g., [3,32,33,100]). Interestingly, the selected period of 50 days, out of the candidate 10–90 days evaluated, corresponds directly with the time lag of some frequently used indices that have been found to be related to fire activity, such as Drought Code (*DC*) (with a time lag of 52 days) ([35,99,101]), 1000 h dead fuel moisture ([17,18]), or 2-month *SPI* [36]. Furthermore, some of the shorter time lag widely used fire danger indices that have been frequently related to fire activity such as *FWI* (e.g., [31,37,102]) or *ERC* (e.g., [4,8]) integrate those longer time lag codes into their weighted calculation (e.g., [2,15]). For example, the 52-day time lag index Drought Code is weighted with 15-day *DMC* in the *FWI* calculation through its contribution to the Build-Up Index (*BUI*) [15]. *BUI* and *DC* are commonly used indicators of potential fuel available for surface fuel consumption, allowing fire managers to evaluate the difficulty in finally extinguishing all areas where the fire is smoldering (e.g., [15]). The index *ERC* is generally calculated for fuel model G (e.g., [2,4,8]), which, owing to a heavy weighting of large dead fuels (100 and 1000 h), is mainly driven by weather conditions during the previous 1.5 months [2].

Beyond a potential influence on coarse dead fuel moisture or on deep soil layers, it is likely that the apparent advantage of the selection of a 50-day period might be related to live fuel moisture dynamics, which are known to influence fire occurrence and spread (e.g., [39,43,103]), although its mathematical contribution to fire modeling remains still as an open research question (e.g., [38,42,43]). In this sense, unlike short-term fire danger indices and dead fuel moisture codes that are driven mainly by short-term weather conditions, live fuel moisture depends not just on recent hydrometeorology [38]. Accumulated vegetation stress is also driven by dynamic and non-linear interactions between weather conditions, soil

properties, and plant physiological processes (e.g., [40,42,43]). Plant responses to drought and dry mass changes associated with phenology are particularly critical ([38,42,104]).

This study offers useful information for a hybrid index that, unlike studies that have focused on weather-driven fire danger indices only (e.g., [22,23,37]]) or only on remotely sensed estimates of live fuel moisture (e.g., [39,43,46]), combines both moisture components. This is, to our best knowledge, the first study analyzing several accumulation periods from 10 to 90 days for *FPI*, one of the few operational fire danger indices integrating both weather-driven dead fuel moisture and remotely sensed live fuel moisture estimates.

Other studies have found the benefit of using time lags beyond 2 months, such as the Monte Alegre formula [105,106], the Telicyn Logarithmic Index [107], and the Nesterov index [108], that use the consecutive number of days without rainfall at longer time periods. Also, the *Risco do Fogo* index, which considers precipitation over a period of 120 days [109]. However, we did not observe gains using longer accumulation times in our study, with correlations generally decreasing for the 90-day interval period.

In addition, our observed non-linearity in the relationship between accumulated *FDI* and fire activity from our study supports similar observations for *FPI* in Europe by [52]. This response has also been observed for other fire weather indices (e.g., [2,102,110]) or for live fuel moisture (e.g., [103]). Consequently, as stated by Koh et al. [111], the common practice of using fire weather indices directly as a proxy for wildfire activity, without a non-linear region-specific calibration to observed fire data, can have limitations in predicting fire occurrence.

This furthermore highlights the need to develop regionally specific calibrated models to convert fire weather indices into estimates of fire activity (e.g., [8,51]). Our high variability in the coefficients to predict fire occurrence might be related to the ample variations in ecosystem types, climate, and human factors between the analyzed states. This corroborates previous observations of variations in such relationships between different regions (e.g., [23,102,112]).

The observed best performance ($R^2$ of >0.6–0.75) in larger states with higher fire activity such as Chihuahua, Ciudad de Mexico, Jalisco, Michoacan, or the State of Mexico, seems to support similar previous observations of stronger relationships at larger areas of study [105]. In general, the range of $R^2$ for the percentile regression models from our study was similar to that found in studies predicting the number of fires in other regions, such as the range of $R^2$ of 0.26–0.46 by [113] or $R^2$ of 0.37–0.60 by [114].

Our observed lower relationship between fuel dryness and fire activity in some of the drier states (e.g., Southern Baja California, Aguascalientes, Sonora, San Luis Potosi, Queretaro, or Guanajuato) agrees with previous studies reporting weaker fuel be related to the observations from previous studies that have similarly documented weaker fuel dryness–fire activity correlations for drier climate regions (e.g., [23,31,37,105]). This might support the hypothesis of varying constraints of fire occurrence (e.g., [115,116]). In addition, weaker weather-fire relationships may arise in regions where fire occurrence is strongly determined by episodic wind-driven fires, such as in Baja California, where Santa Ana winds are known to influence fire activity [117]. Current ongoing research aimed at developing a windy *FPI* for Mexico, similar to what recently developed for a revised *FPI* index including wind in the USA [118] or other countries [119] might contribute to improving our capacity to predict fire occurrence in future studies. This could provide information at finer temporal scales than the periods of 10 days evaluated in the current study. In this regard, different time lags of fuel moisture than the ones selected for 10-day fire occurrence here, together with the consideration of wind, might be useful for future analyses aiming at predicting daily fire spread. In addition, future studies could expand the period of study, based on the future availability of *FDI* data.

Beyond observing the effect of fuel dryness in promoting fire activity, for many of the analyzed states, a large effect of temporal autocorrelation was also documented. For example, in spite of lower observed accumulated fuel dryness in 2015 for the state of Oaxaca (Figure 4c), the relatively large fire observed activity could be generally predicted, even under wetter conditions, because of the consideration of autocorrelation terms. This

approach predicts a higher number of fires as a response to observed fire activity from the previous 10 days (Figure 4c). In this sense, many of the states with the largest *c* coefficient in the predictive equation to account for temporal autocorrelation of previous fire activity (Table 1), such as Oaxaca, Chiapas, Michoacan, Jalisco, or Nayarit, correspond to areas in the center and southeast of Mexico where fires are most likely a result of the spread of frequent agricultural burns (e.g., [120,121]). These observations support the notion that anthropogenic factors can effectively mask or have an influence beyond weather–fire relationships in some regions due to extensive and regular intentional human ignitions where conversion to agricultural lands and greater land fragmentation occurs (e.g., [23,61–63]).

Our use of an autoregressive term of 1 lag, allowing us to forecast the number of fires based on the observed number from the previous 10 days, agrees with the observations of [65] who found a temporal correlation of up to 11 days to predict daily arson ignition counts in Florida. Nevertheless, our study, unlike the univariate autoregressive approach of Prestemon et al. [66] or other authors [67,68], or unlike the majority of literature that have used fire danger indices only (e.g., [23,37,102]), included a combination of both fuel dryness and autoregressive terms of previous fire activity, to forecast fire activity of the next 10 days.

In our study, on the one hand, autocorrelation alone explained a relatively large variability of the observed number of fires for the majority of the evaluated states, as noted by a correlation coefficient (*c*, Table 1) of up to 0.8, being >0.5 for 20 of the analyzed states. On the other hand, adding fuel dryness in addition to autocorrelation improved the correlation for the large majority of the analyzed states (Table 1). For example, in the Campeche state, considering only autocorrelation resulted in an $R^2$ (squared correlation coefficient of 0.40) of 0.16 (Table 1). Conversely, including fuel dryness in addition to autocorrelation increased the $R^2$ for this state to 0.45 (Table 1). Furthermore, using previous fires only, would not allow us to anticipate either (1) the beginning of the fire season (when previous days show few or no fire activity), nor (2) sudden peaks of fire activity when weather conditions aggravate, after previous days of moderate fire activity. This can be observed in the plots of predicted fire activity (Figure 1). For example, for Chihuahua (Figure 4a), at the beginning of the fire season of the year 2012, the previously observed number of fires from the previous month of December 2011 was 0. Using only an extrapolation of the previous fires, one would assume 0 fires for the first 10 days of January, underestimating the observed start of the fire season on this date. Instead, because the model provides a conservative (percentile-fitted) estimate of fire activity based on accumulated fuel dryness, the observed number at the beginning of the fire season is not underestimated (Figure 4a). Also, the sudden increases in fire activity at the end of January 2012 and the start of March 2012 in this state (Figure 4a), which would have otherwise been underestimated based on previously observed fire activity, were successfully anticipated in the predictions because of the consideration of accumulated fuel dryness. This suggests that considering a percentile fit of fire activity against fire weather can provide conservative estimates of fire activity, anticipating sudden peaks of fire occurrence in dry seasons before they have occurred (Figure 4), as desired for a safe fire hazard decision support tool.

Autoregressive terms were also used to forecast the accumulated fuel dryness of the next 10 days. Our approach agrees with the results of Huesca et al. [53] who used autoregressive models of the previous 2 lags to predict *FPI* in Spain. Although the autoregressive models developed in this study showed good skill ($R^2$ of 0.989 to 0.813) in forecasting the accumulated fuel dryness of the next 10 days, future studies could further explore the use of weather forecasts (e.g., [4,122]). Such approaches could be valuable to predict fire occurrence under potentially changing climate conditions [9–12], including expanded fire seasons, or changes in the timing of precipitation, that might be better captured with such more detailed weather forecasts.

The current study aimed at forecasting the total number of fires by state. Nevertheless, considering that large fires can represent a large fraction of the fire suppression budget (e.g., [5,123–125]), future studies could aim at predicting a number of large fires (e.g., [8]).

Furthermore, beyond developing temporal forecasts of large fire activity based on the average (or percentile) fuel dryness value by state, more detailed spatio-temporal approaches to predict fire occurrence, such as those demonstrated by Preisler et al. [8,19]), should be explored in future studies. Such approaches could allow both mapping fire or large fire occurrence probability and simultaneously estimating the number of fires or the number of large fires for a particular region and period of time by summing the estimated probability values of individual voxels (e.g., [51]). This would further contribute to support decision-making not only between but also within states, potentially improving fire suppression and fire management planning.

## 5. Conclusions

The main conclusions of the study can be summarized as:

(1) This study evaluated for the first time the effect of different accumulation time periods on the capability of a modified version of the *FPI* fire danger index to forecast the number of fires.

(2) Our results suggest that a period of 50 days, provided the best results to forecast fire activity in a variety of geographical areas with different ecosystems and climates in Mexico. These results indicate a potential effect of the selected time period in capturing live fuel moisture dynamics effects in fire occurrence for the study area analyzed, that could be tested for accumulated *FPI* in other research areas using the methodology presented here.

(3) In addition, the use of autoregressive terms, in combination with the accumulated fuel dryness, reveals its usefulness for predicting fire activity for several states. This approach could be tested elsewhere based on *FPI* or other fire indices to forecast fire activity.

(4) Finally, autoregressive models showed good performance in forecasting Accumulated Fuel Dryness (*AcFDI*) based on previously observed *AcFDI* values, allowing the development of forecasts of the expected number of fires by state for the next 10 days.

(5) Future studies might enhance the potential of these initial models by exploring weather forecasts of both predicted fuel dryness and wind. Furthermore, spatio-temporal approaches should be also tested to further support fire management planning at different time and spatial scales.

**Supplementary Materials:** The following supporting information can be downloaded at: https://www.mdpi.com/article/10.3390/f15010042/s1, Figures S1–S5: Observed and predicted number of fires per dekad for the states of Chiapas, Ciudad de Mexico, Durango and Guerrero (S1), Hidalgo, Jalisco, Michoacan and Morelos (S2), Puebla, Nayarit, Queretaro and San Luis Potosi (S3), Sinaloa, Sonora, Tlaxcala, and Veracruz (S4) and Zacatecas, Coahuila, Campeche and Yucatan (S5). Figures S6–S10: Observed and predicted accumulated fuel dryness FDI50 for the states of Chiapas, Ciudad de Mexico, Durango and Guerrero (S6), Hidalgo, Jalisco, Michoacan and Morelos (S7), Puebla, Nayarit, Queretaro and San Luis Potosi (S8), Sinaloa, Sonora, Tlaxcala, and Veracruz (S9) and Zacatecas, Coahuila, Campeche and Yucatan (S10).

**Author Contributions:** D.J.V.-N., P.-M.L.-S. and J.B.-R. performed the statistical analysis; J.B.-R. programmed the code for the daily Fuel Dryness Index automated calculation and for the extraction of FDI values to the daily fire hotspots; R.E.B. provided the Fire Potential Index algorithm, upon which much of this research is based; M.I.C.-L., M.C. and R.R. calculated daily H100 and 10-day NDVI composites from satellite information from CONABIO; D.J.V.-N., writing—original draft preparation; J.J.C.-R., M.P.-G. and E.A.-C., writing—review and editing. All authors have read and agreed to the published version of the manuscript.

**Funding:** Funding for this study was provided by CONAFOR/CONACYT Project "CO-2018-2-A3-S-131553, Reforzamiento al Sistema Nacional de Predicción de Peligro de Incendios Forestales de México para el pronóstico de conglomerados y área quemada (2019–2022)", for the enhancement of the Forest Fire Danger Prediction System of Mexico to map and forecast active fire perimeters and burned area, and by the project CONAFOR/CONACYT Project C0-3-2014 "Development of a Forest Fire Danger Prediction System for Mexico (2015–2017)" for the development of a Forest Fire Danger Prediction System for Mexico, funded by the Sectorial Fund for forest research, development and technological innovation "Fondo Sectorial para la investigación, el desarrollo y la innovación tecnológica forestal".

**Data Availability Statement:** Fire suppression registers from CONAFOR can be downloaded from the section "Incendios" of the Forest Fire Danger Forecast System of Mexico, "Sistema de Predicción de Peligro de Incendios Forestales de México" (SPPIF): http://forestales.ujed.mx/incendios2/Fuel (accessed on 20 December 2023) dryness indexes can be visualized in SPPIF and are available upon request to the authors.

**Acknowledgments:** We want to thank CONAFOR personnel for providing the fire suppression registers analyzed in the study. We also want to thank CONABIO's personnel for providing us access to the satellite daily data of fire hotspots, 10-day NDVI composites, and daily dead fuel moisture images for Mexico for the period of study.

**Conflicts of Interest:** The authors declare no conflicts of interest. The founding sponsors had no role in the design of the study; in the collection, analyses, or interpretation of data; in the writing of the manuscript, and in the decision to publish the results.

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
