# Peer review of "Autoregressive Forecasting of the Number of Forest Fires Using an Accumulated MODIS-Based Fuel Dryness Index"

_forests, doi:10.3390/f15010042_

Round 1

Reviewer 1 Report

Comments and Suggestions for Authors

I would like to get just a few responses from the authors concerning the following issues:

1. Line 120: Please give more information about this catalogue. Can you provide a reference to the resource? The reference is required.

2. Lines 123–128: It would be better to display such data on a map of the territory.

3. Eq. 4: I don’t understand the meaning of the numerical coefficients 30, 255, 125. Why do NDVI have values of 125 and 250 for Mexico? This indicator has a range of values 0.0–1.0. Please, clarify here.

4. Eq. 6 and lines 182 and Table 1: How were these coefficients determined? Please clarify. What determines the high variability of these coefficients?

5. Fig. 2: Perhaps this data would look more interesting in map form?

6. Fig. 3: Can the model self-update for a subsequent time interval? Why is there no forecast for the current decade or for the next season?

7. Lines 258–263: It seems to me that this description is more appropriate in the Methods section.

8. The methods do not describe how the data were generalized across the state. Methods (lines 130–166) provide equations using satellite (1 km resolution) estimates of vegetation and moisture conditions. However, it is necessary to explain how they are processed in order to obtain a graph for each territory (Fig. 4).

9. It can be assumed, that a simple extrapolation of fires number over the past 10 days to future period will be not less reliable. The question is: what is the advantage of your more complex predictive model?

Reviewer 2 Report

Comments and Suggestions for Authors

In recent decades, we have seen forest fires turn from a local disaster into catastrophes with global negative economic and environmental consequences. Fire hazard monitoring, organization of protection and restoration of forest resources have acquired state status. In these conditions, scientific research on forest fire forecasting is extremely relevant. The successful results of such studies will make it possible to carry out more effective organization of fire-fighting measures and, if not prevent, then at least partially mitigate the damage. The reviewed article presents the results of mathematical modeling and forecasting of the number of forest fires based on the hybrid Fuel Dryness Index (FDI) and the observed number of fires over the previous 10-90-day time periods using the example of 32 Mexican states for 2011-2015. The hybrid fuel dryness index is a characteristic of two types of humidity: vegetation based on NDVI and residual environmental humidity based on H100 (observed 100h dead fuel moisture) according to MODIS satellite data with a spatial resolution of 1 km. As a result of the study of the correlation of the number of fires with the accumulated value of the AcFDI index (characteristic of accumulated vegetation stress due to drought for periods from 10 to 90 days), a 50-day period with the highest coefficient of determination r2>0.4 values was established for 19 states. In states where correlation is weak or absent, the frequency of fires is reduced due to higher humidity, wind and other unidentified causes. A period of 50 days is consistent with the time intervals of some commonly used indexes that are associated with fire activity, such as the Drought Code (DC) (with a time lag of 52 days), residual fuel humidity of 1000 h (H1000), 2-month SPI. The authors of the article also built a second-order autoregressive model for FDI50, with an incredibly high correlation of predicted and calculated values coefficient of determination r2>0.95 for 29 states. The proposed method of combining the characteristics of vegetation dryness and the environment during a long pre-fire period is of undoubted interest and has great potential for regions with other climatic and natural-geographical conditions.

Remarks:

1. There is a lack of information about the study area, vegetation characteristics, and the relevance of the forecast on the number and scale of fires for Mexico.

2. When describing the data used, there is no mention of the problem with clouds. Is the data for cloud time intervals ignored or somehow replenished?

3. Is there an error in the formula 4: Rmax (must be LRmax)? Why do the maximum and minimum NDVI values for Mexico City take values greater than 1? Or do you mean accumulated values or scaled ones?

4. Error in the name of the vertical axis FD99 (must be FDI99) on fig.2. Maybe Figure 2 would look better not as a histogram, but as a map with colored states in accordance with the values of FDI99?

5. In Table 2, it is better to reduce the number of significant digits after the decimal point of the model coefficients a0, a1, a2.

Reviewer 3 Report

Comments and Suggestions for Authors

Dear authors!

I think your article is in the scope of Forests journal.

But at present time has some disadvantages.

Title

I think Title is overloaded and regionally emphasized at present time.

I suggest to reduce and rearrange the title:

Autoregressive Forecasting the Number of Forest Fires Using an Accumulated MODIS-based Fuel Dryness Index: Case Study

Abstract

You shoud not use "dekad". This is a mistake. Decade is ten years in English.

Keywords

I think you should not use term "risk" in suggested set of keywords.

Your articled is targeted to forest fire danger (hazard) forecasting.

Introduction

You should insert sentence "The aim of this study is to develop ...".

Then you can write objectives instead of goals.

Your literature review is limited by weather caused forest fire danger.

But I think you should provide some brief information on fire reasons like lightning and human activity. Forest fires occurrs in fire weather conditions but forest fire can not be initiated without ignition caused by lightning or human activity. 

Materials and Methods

Please, give some support to solid the suggested Forest fire number non-linear dependence.

Please, briefly describe the mathematical basis of ARIMA analysis within this section. I think this is obligatory for research article. What software was used to provide ARIMA analysis? Is it your own software or any early issued software?

Results and Discussion

Please, clarify limitations of your study and future research descriptions.

Conclusions

I suggest to make numbered set of 4-6 key findings with corresponding conclusions.

References

Please, add some information on forest fire occurrence reasons like lightning and human activity.

Supplementary materials

Please, explain negative numbers in Supplementary materials, for example, for Hidalgo results (page 2, Figure S1.2)

Comments on the Quality of English Language

No comments

Round 2

Reviewer 1 Report

Comments and Suggestions for Authors

Author Response

Thank you for a constructive review.
Please see the attachment for the final minor comments, now fully responded in the final manuscript. 

Reviewer 3 Report

Comments and Suggestions for Authors

Dear authors!

Thank you for your revision.

I have accepted all your improvements and explanations.

Comments on the Quality of English Language

No comment

Author Response

Thank you for your constructive revision, all your comments were included and helped enrich the manuscript.